# Applications of Cyclodextrin-Based Drug Delivery Systems in Inflammation-Related Diseases

**DOI:** 10.3390/pharmaceutics17030378

**Published:** 2025-03-17

**Authors:** Zelan Dai, Huijuan Yang, Peng Yin, Xingkang Liu, Ling Zhang, Youwei Dou, Shibo Sun

**Affiliations:** 1Department of Pulmonary and Critical Care Medicine, First Affiliated Hospital, Kunming Medical University, Kunming 650032, China; dzl621222@163.com; 2Department VII of Biological Products, Institute of Medical Biology, Chinese Academy of Medical Sciences & Peking Union Medical College, Kunming 650031, China; huijuanyang@imbcams.com.cn (H.Y.); yp199604@163.com (P.Y.); 14787809648@163.com (X.L.); zhangling951205@163.com (L.Z.); youwei_dou@126.com (Y.D.)

**Keywords:** cyclodextrin, drug delivery system, nano-delivery, inflammation

## Abstract

Currently, inflammation diseases are one of the leading causes of mortality worldwide. The therapeutic drugs for inflammation are mainly steroidal and non-steroidal anti-inflammatory drugs. However, the use of these anti-inflammatory drugs over a prolonged period is prone to causing serious side effects. Accordingly, it is particularly critical to design an intelligent target-specific drug delivery system to control the release of drugs in order to mitigate the side effects of anti-inflammatory drugs without limiting their activity. Meanwhile, cyclodextrin-based nano-delivery systems have garnered significant attention in contemporary pharmaceutical research owing to their capacity to enhance drug bioavailability, enable site-specific targeted accumulation, prolong the systemic circulation duration, facilitate synergistic therapeutic outcomes, and exhibit superior biocompatibility profiles. It is worth noting that cyclodextrin-based drug delivery systems show great potential in inflammation-related diseases. However, few studies have systematically reviewed their design strategies and application advancements. Here, we summarize the structural and chemical modification strategies of cyclodextrins, as well as cyclodextrin-based drug delivery systems and their applications in inflammation-related diseases. In summary, the aim is to provide a bit of insight into the development of cyclodextrin-based drug delivery systems for inflammation-related diseases.

## 1. Introduction

Inflammation is a physiological process by which the immune system responds to harmful stimuli such as pathogens, damaged cells, and toxic compounds [1,2]. Normally, the inflammatory response, which is aimed at repairing damaged tissues and removing harmful factors such as pathogens [3], is beneficial to the body. However, uncontrolled inflammation is implicated in many diseases [1,4]. At present, the incidence of inflammatory diseases is on the rise, and inflammatory diseases have become one of the leading causes of mortality worldwide [5,6,7]. Inflammation increases the risk of many diseases, such as cancer [8], non-alcoholic fatty liver [9], metabolic syndrome [10], type 2 diabetes [11], rheumatoid arthritis, inflammatory bowel disease, atherosclerosis, allergies, asthma, and psoriasis. The current therapeutic drugs for inflammation are mainly steroidal and non-steroidal anti-inflammatory drugs (NSAIDs) [12]. Due to the systemic side effects of steroidal anti-inflammatory drugs, NSAIDs tend to be the preferred medications for the long-term treatment of inflammation, of which ibuprofen, indomethacin, and diclofenac are all commonly used [13,14]. However, the use of these anti-inflammatory drugs over a prolonged period is prone to causing serious side effects, such as gastrointestinal damage, cardiovascular toxicity, and renal abnormalities [12,15]. Accordingly, in order to mitigate the side effects of anti-inflammatory drugs without limiting their activity, it is particularly critical to design an intelligent target-specific drug delivery system to control the release of drugs. In this system, the precise distribution of drugs enables quantitative release in terms of time, space, and dose control, achieving optimal biological efficacy at the lowest dose and minimizing side effects [16].

As a traditional drug carrier, cyclodextrins (CDs) are worth building due to their unique annular cavity structure, modified surface hydroxyl groups, good biocompatibility, inclusion complex ability, and water solubility [17,18,19,20]. Cyclodextrin-based nano-delivery systems have garnered significant attention in contemporary pharmaceutical research owing to their capacity to enhance drug bioavailability, enable site-specific targeted accumulation, prolong the systemic circulation duration, facilitate synergistic therapeutic outcomes, and exhibit superior biocompatibility profiles [21,22,23]. It is worth noting that cyclodextrin-based nano-delivery systems can achieve the targeted delivery and controlled release of drugs in the treatment of inflammation-related diseases, thus ameliorating inflammation. For example, Yuan Wang et al. developed a cyclodextrin-based nano-delivery system with pH/ROS dual responsibility for inflammatory bowel disease [24]. Cyclodextrin-based drug delivery systems show great potential in inflammation-related diseases. However, few studies have systematically reviewed their design strategies and application advancements.

In this review, we summarize the structure and chemical modification of cyclodextrins. Second, the applications of cyclodextrin-based drug delivery systems in inflammation-related diseases are summarized. Finally, the current obstacles and future directions of cyclodextrin-based delivery systems are discussed.

## 2. Structure and Properties of Cyclodextrins

Cyclodextrins (CDs) are produced by the degradation of starch under the action of CD glucosyltransferase, which is a class of cyclic oligosaccharides formed by five or more α-D-glucopyranose units linked by α-(1,4) glucoside bonds. Common cyclodextrins are α-cyclodextrin (α-CD), β-cyclodextrin (β-CD), and γ-cyclodextrin (γ-CD) containing six, seven, and eight glucose subunits, respectively [25,26,27]. β-CD is widely used in pharmaceutical excipients because of its suitable cavity size, effective drug load, and relatively low cost [28]. Cyclodextrin is a physically and chemically stable macromolecule with a truncated cone or torus structure with a hydrophilic external structure and a hydrophobic internal cavity [29,30,31,32]. Such a structural feature presents a unique dual property that interacts with both lipophilic and hydrophilic compounds [33]. Importantly, the hydrophobic internal cavity can trap or encapsulate drugs (guests), thus forming non-covalently bound host–guest inclusion complexes to improve the physical, chemical, and biological properties of the guest molecules [32,34,35,36]. Figure 1 shows the chemical structure of cyclodextrin.

## 3. Chemical Modification of Cyclodextrins

In order to broaden the range of applications of CDs in drug delivery systems, various CD derivatives have emerged along with the continuous development of novel technologies. Desired functional groups can be easily and precisely introduced into the predetermined sites through a wide variety of chemical modifications, thus expanding their range of applications. For example, modified β-CD has better solubility than unmodified β-CD [37]. Hydroxypropyl-, sulfobutylether-, and carboxymethyl-type β-CDs are multifunctional derivatives that have been used to improve the solubility, stability, and physical properties of CD–guest complexes [38,39]. Among them, hydroxypropyl-β-cyclodextrin (HP-β-CD) and sulfobutylether-β-CD (SBE-β-CD) have been studied and registered [37,40]. Some other methylated derivatives and maltosyl β-CD (Ma-β-CD) have also been used in production. Hydroxypropyl-γ-CD has lower aggregation than unmodified γ-CD [41]. In addition to these hydrophilic derivatives, some amphiphilic derivatives have also been studied [42]. Due to the special structure of CDs, it is possible to produce polymers with different structures and functions through chemical reactions, such as CD-centered star polymers, CD-capped polymers, and CD-pendant polymers [43].

## 4. Nano-Delivery Systems Based on Cyclodextrins

Cyclodextrins (CDs), a class of classic drug carriers, have demonstrated their ability in nano-delivery systems due to their unique structural advantages. The hydrophobic cavity composed of a cyclic oligosaccharide structure can realize supramolecular self-assembly through host–guest interactions, and the abundant hydroxyl groups on the surface provide a high degree of tunability for molecular modification. Based on these characteristics, researchers have successfully constructed a cyclodextrin-based nanodrug delivery system by combining cyclodextrin with nanomaterials with targeting properties. According to the composition and characteristics of nanosystems, CD-based nano-delivery platforms are mainly classified into polymeric nanosystems based on cyclodextrins, graphene derivative delivery systems modified with cyclodextrins, cyclodextrin-based inorganic nanoparticle delivery systems, cyclodextrin–liposome systems, and other nano-delivery systems based on cyclodextrins. Interestingly, an intelligent stimuli-responsive drug delivery system was fabricated by incorporating some reactive functions and was capable of controlling the release of drugs according to different stimuli [44,45,46,47] that included pH, temperature, redox, enzymes, light, and magnetism [48]. Table 1 summarizes important data on cyclodextrin-based delivery systems.

### 4.1. Polymeric Nanosystems Based on Cyclodextrins

CD-based polymeric nanoparticles exhibit unique advantages in the field of drug delivery [67,68]. However, the nanoparticles have inherent limitations, such as low encapsulation rates and drug loading. Nanoparticles aided by CDs yield a novel drug delivery system with the benefits of both components: the CDs offer improved water solubility and drug loading, whereas the NPs provide targeted drug delivery. The delivery systems have cyclodextrin casting outer shells, while the core of the polymeric nanoparticles is composed of natural or synthetic polymer. The drugs can either be loaded into the polymer nanoparticle core or be conjugated with cyclodextrin in the outer shell [19,67]. Many studies have shown the importance of CD-based polymer nanoparticle systems for realizing stimuli-responsive drug delivery systems. An efficient delivery system was developed by combining CDs and polymer nanoparticles (PNPs). CDs can improve the efficiency of drug loading, and PNPs can be utilized to synergize this intelligent delivery system in response to endogenous stimuli (such as pH, enzymes, and temperature) or exogenous stimuli (such as magnetism, light, and ultrasound), thereby accelerating the release of drugs [69]. However, this composite system still faces some limitations in practice. For example, the preparation of polymer nanoparticles is a complex process that is highly sensitive to solvent selection, PH, and temperature [70,71].

Supramolecular self-assembled nanocarriers based on CDs also belong to polymeric nanosystems, and their performance can be optimized by molecular functional strategies. Studies have shown that the structural modification of β-CD with different functional groups (such as amantadane, azobenzene, etc.) or polymers (such as PEG, PEI, etc.) can improve the solubility of the therapeutic molecules and the stability of the inclusion complexes and regulate the interface affinity of the macromolecular self-assembly systems in the biological environment, thus overcoming the excessive aggregation problems of macromolecular self-assembly systems [49,72]. For example, amantadine and amphiphilic β-CD formed a stable molecular aggregate that was used to deliver HPTS (8-hydroxypyren-1,3,6-trisulfonic acid). After 360 min, the release rate of HPTS encapsulated in supramolecular nanoparticles in water was reported to be 8-fold lower than that of free HTPS (80%) [49]. In addition, a nanocarrier was constructed for targeted delivery and combination chemotherapy, utilizing the target specificity of aptamers (oligonucleotides) and the self-assembling capabilities of CD-based supramolecules. The results showed that self-assembling nanocarriers based on CDs provide an efficient platform for targeted drug delivery systems with synergistic antitumor activity [50]. However, CD-based supramolecular self-assembled nanosystems still face technical bottlenecks in their applications. First, there exists instability in the structure of the host–guest complex under acid–base conditions, which leads to some covalent/non-covalent assemblies prone to dissociation. Second, supramolecular self-assembled nanosystems based on CDs are extremely sensitive to the parameters in the process (solvent polarity, concentration gradient, stirring speed, etc.). However, the vulnerability can be weakened by CD–polymer conjugation, which has better loading rate and stability [73,74].

### 4.2. Graphene Derivative Delivery Systems Modified with Cyclodextrins

In drug delivery systems, graphene-derived nanoparticles were introduced to increase drug loading and control drug release by stimuli (e.g., alternating magnetic fields, changing pH or temperature) [75]. Among them, graphene oxide (GO) has become a research hotspot in the field of nanomedicine delivery due to its high specific surface area and abundant surface functional groups [76]. CD-modified graphene oxide (GO) nanomedicine delivery systems take advantage of the cavity structure of CD and the stimuli-responsive release ability of GO. The delivery systems have a high drug loading capacity and loading efficiency, enabling targeted delivery and controlled release. Graphene oxide (GO) has been a hot topic of research [76]. Interestingly, a CD-modified nanocarrier based on GO was synthesized for cancer therapy using L-phenylalanine as a link, and the load efficiency and load capacity of this system for DOX were 78.7% and 85%, respectively. After 72 h, only 12% of DOX was slowly released at pH 7.4, while up to 40% of DOX was released at pH 5.3 under acidic conditions [51]. In addition, a supramolecular hydrogel consisting of GO and α-CD was designed. This supramolecular hydrogel increased drug loading through host–guest interactions and released the anticancer drug 5-fluorouracil in response to near-infrared light (NIR), temperature, and pH. In this system, GO was used as a core material for additional crosslinking to confer thermal stability. Meanwhile, GO absorbed near-infrared light and converted part of the heat into local heat to trigger the gel-to-sol transition [52]. However, there are still some limitations in the application of CD-modified graphene derivative delivery systems. First, although the cytotoxicity possessed by GO can partially be reduced by CD modification, the long-term biosafety of this composite system is not yet clear [77,78]. In addition, most of the existing methods for the preparation of CD-modified GO rely on sophisticated chemical reactions, which makes them difficult to convert to large-scale production [79].

### 4.3. Cyclodextrin-Based Inorganic Nanoparticle Systems

Cyclodextrin-based inorganic nanoparticle delivery systems have attracted much attention due to their unique advantages in the field of targeted drug delivery and sustained release. In this system, common inorganic nanomaterials mainly include mesoporous silica nanoparticles, plasmonic nanoparticles, magnetic nanoparticles, and quantum dots. Among them, mesoporous silica nanoparticles exhibit significant advantages in targeted delivery due to their internal pores with a large surface area and tunable mesoporous structure [80]. Since mesoporous silica nanoparticles have no ability to respond to stimuli, they must be chemically modified using polymers or supramolecular structures to control the release of drugs. On the one hand, derivatives of CDs were used as gatekeepers to prevent the premature release of drugs. Furthermore, CDs were loaded with drugs into mesoporous silica matrices and responded to external stimulus signals through host–guest interactions [81]. On the other hand, derivatives of CDs contributed to the synthesis of these nanoparticles as reductants and stabilizers and increased the loading capacity of the drugs [82].

However, despite the potential of this system in terms of drug loading, biocompatibility, and targeting, the system has several limitations that restrict its clinical translation and application. Firstly, mesoporous silica nanoparticles can realize drug loading by means of physical adsorption or chemical bonding through the mesoporous structure, but their structure is unstable in complex physiological environments (such as pH fluctuations) in vivo [83]. Secondly, mesoporous silica nanoparticles are easily adsorbed by plasma proteins to form a “protein corona” following systemic administration, thus obscuring the targeted ligands and altering their biological distribution [84,85]. Furthermore, the biodegradability of mesoporous silica materials has not been fully elucidated in vivo, which may bring some long-term physiological toxicity [86]. What is more, the long-term retention of high doses of mesoporous silica can even cause organ fibrosis [87].

### 4.4. Cyclodextrin–Liposome Systems

Liposomes are small spherical vesicles that are synthesized and are composed of one or more lipid bilayers that enclose the inner aqueous space [88]. The structure makes liposomes suitable for encapsulating hydrophobic, hydrophilic, and amphiphilic substances [89]. Different strategies have been developed for the binding of CDs to liposomes, one of which is called “drug-in CD-in liposome systems”. The inclusion complexes are encapsulated in the aqueous nuclei of different liposomes. The dual-loading nano-delivery system increases the solubility of the active pharmaceutical ingredient and the stability of the vesicles. Compared to single-loading mode, this dual-loading technology enables either the rapid or prolonged release of the active pharmaceutical ingredient [90,91]. Drug-in CD-in liposome formulations have been shown to improve the encapsulation efficiency and loading ratio [92].

However, there may be some issues that need to be considered regarding composite delivery systems of CDs and liposomes. First, drug-in CD-in liposome formulations should take into account the effect of CDs on the membranes of liposomes [93]. In short, CD molecules present in the aqueous phase space of the liposome will interact with phospholipids on the liposome membrane, thus affecting the stability of CDs and changing their fluidity and permeability [94,95]. Furthermore, lipids exhibit hemolytic activity, as evidenced by in vitro experimental studies [96]. Accordingly, the potential toxicity associated with the CD–liposome system needs to be investigated.

### 4.5. Other Nano-Delivery Systems Based on Cyclodextrins

In addition to the nanomaterials mentioned above, CDs or their derivatives can also interact with metal–organic frameworks (MOFs) [97], Janus nanoparticles [98], nanofibers [99], or nanomicelles to build novel drug delivery systems.

## 5. Applications of Cyclodextrin-Based Drug Delivery Systems in Inflammation-Related Diseases

CD-based drug delivery systems have been used for the treatment of a variety of inflammation-related diseases, including respiratory diseases, inflammatory bowel diseases, vascular diseases, osteoarthritis, inflammatory eye diseases, neurodegenerative diseases, etc. Figure 2 shows the assembly process of a cyclodextrin-based nano-delivery system. Figure 3 shows the applications of cyclodextrin-based drug delivery systems in inflammation-related diseases.

### 5.1. Applications of Cyclodextrin-Based Delivery Systems in Respiratory Diseases

Drug delivery systems based on CDs are widely used for the treatment of respiratory diseases. For example, the inclusion cocrystal of asiatic acid/γCD reduced the levels of pro-inflammatory cytokines in the lungs and bronchoalveolar lavage fluid to execute the anti-inflammatory effect in vivo in mice with acute lung injury (ALI) [53]. Pulmonary fibrosis is an interstitial lung disease characterized by progressive and often fatal dyspnea. An inclusion complex consisting of a traditional Chinese medicine ingredient and hydroxypropyl-β-CD was used to treat pulmonary fibrosis via inhalation administration. Animal studies showed that the inclusion complex attenuated inflammation and fibrosis in the lungs [54]. Similarly, γ-CD-based metal–organic frameworks delivered drugs to the lungs to treat fibrotic interstitial lung disease [55]. Additionally, an inclusion complex composed of Chinese herbal extracts and β-CD was used to treat Staphylococcus aureus pneumonia, effectively inhibiting bacteria and mitigating inflammation [56]. Furthermore, a cyro-shocked macrophage-based drug delivery system was prepared to treat acute pneumonia via the supramolecular conjugation of CD-modified “zombie” macrophages with nanocarriers functionalized with amantadane (ADA). The “zombie” macrophages carried nanocarriers loaded with quercetin, which effectively alleviated inflammation in the lungs of mice [100]. In contrast, the administration of curcumin–cyclodextrin complexes resulted in a significant decrease in angiotensin converting enzyme 2 and activator of transcription 3 in mice following pneumonia, suggesting the potential of curcumin–cyclodextrin complexes in the treatment of COVID-19 [101].

### 5.2. Applications of Cyclodextrin-Based Delivery Systems in Inflammatory Joint Diseases

Poly-CD-based nanoassemblies loaded with the non-steroidal anti-inflammatory drug diclofenac sodium (DCF) and crosslinked with a fluorescent probe (adamantanyl–rhodamine conjugate) by supramolecular interaction were developed to control inflammation in bone and joint diseases. Ultimately, the nanoassemblies inhibited the production of inflammatory factors in human bone marrow mesenchymal stem cells, and the CD-based nanomaterial may become a drug carrier for the treatment of bone and joint diseases [57]. Specifically, β-CD modified with polyethylene glycol (PEG) and aspartate hexapeptide (Asp6) was loaded with norfloxacin to treat bone infections. The antimicrobial capacity of norfloxacin was enhanced through a β-CD-based delivery system, thereby reducing inflammation and promoting bone tissue repair [102]. The sustained release of the drug was achieved by utilizing a carrier consisting of mesoporous silica nanomaterials and β-CD in which anti-osteoarthritis compounds were loaded. It was observed that the delivery system ameliorated both systemic and local inflammatory responses while also slowing the progression of joint degeneration and arthritis in vivo in rats [103]. Additionally, osteoarthritis was treated effectively by utilizing gelatin–glucosamine hydrochloride mixed crosslinked cyclodextrin metal–organic framework composite hydrogels as carriers for delivering anti-inflammatory drugs [104].

### 5.3. Applications of Cyclodextrin-Based Delivery Systems in Inflammatory Bowel Diseases

The oral administration of drugs for ulcerative colitis faces several obstacles. A yeast pellet system loaded with supramolecular nanoparticles encapsulating curcumin was utilized to treat ulcerative colitis. In this supramolecular nano-delivery system, β-CD bound with double-sensitive material formed an inclusion complex with adamantane (ADA) modified by D-mannose via a host–guest interaction and was then loaded with curcumin. Ultimately, the supramolecular nanoparticles were encapsulated in yeast cells. This CD-based supramolecular nanoassembly effectively improved the damaged colon tissue [58]. Furthermore, to effectively treat inflammatory bowel disease, a novel H_2_O_2_-responsive covalent cyclodextrin delivery system was prepared by crosslinking the H_2_O_2_-activated antioxidant prodrug BRAP with cyclodextrin-based metal–organic frameworks, which reduced inflammation in the body and alleviated the worsening of inflammatory bowel diseases [59]. Similarly, β-CD modified by ROS-sensitive material was utilized to load therapeutic drugs, and the inclusion complex was then coated with macrophage membranes. This system was used for the targeted therapy of ulcerative colitis [105]. Likewise, nanosystems containing oxidation-sensitive cyclodextrin (OX-CD) were created to treat inflammatory bowel disease. In this system, OX-CD was loaded with the drugs, and chitosan and pectin with opposite charges were sequentially coated onto the OX-CD to form a nanosystem with dual pH/ROS sensitivity and the capability of targeting inflammation [24]. Interestingly, amphiphilic cationic CDs as carriers delivered TNF-α siRNA to macrophages and mice that were induced with acute colitis; the results suggested that the CD-based delivery systems had some potential to deliver TNF-α siRNA for the treatment of inflammatory bowel diseases [106].

### 5.4. Applications of Cyclodextrin-Based Delivery Systems in Inflammatory Eye Diseases

Glaucoma is an ophthalmic disease that exhibits inflammatory changes to various extents along the eye structure [107]. Poly (2-hydroxyethyl methacrylate) hydrogel containing β-CD (pHEMA/β-CD) contact lenses were used for the ocular administration of puerarin in the treatment of glaucoma. It was shown that pHEMA/β-CD hydrogel contact lenses extended the duration of drug retention in the eye and increased the concentration of the drug in the aqueous humor compared to eye drops in vivo in rabbits [60]. Similarly, the chemical modification of pHEMA hydrogel contact lenses with β-CD was investigated as a sustained release platform for other hydrophobic ocular administrations [61,62].

CD-based drug delivery systems were employed to treat conjunctivitis, which is also an eye disease with inflammation. For example, an inclusion complex composed of an active compound and cyclodextrin was encapsulated in gel in situ. This delivery system enhanced the anti-inflammatory activity and extended the retention time of the active compound in the eye, thus effectively treating conjunctivitis [108]. Furthermore, chitosan/sulfobutylether-β-cyclodextrin-based nanoparticles coated with thiolated hyaluronic acid were used for the ophthalmic administration of indomethacin, which effectively increased the solubility and residential time of the drug in the conjunctiva sac [63].

### 5.5. Applications of Cyclodextrin-Based Delivery Systems in Vascular Diseases

Currently, the pharmacologic treatment of atherosclerosis focuses on cholesterol control and inflammation management. The cyclodextrin-mediated conjugation of macrophages and liposomes was conducted to treat atherosclerosis by means of plaque lysis and inflammation reduction. Derivatives of β-CD and adamantane were used to modify macrophage membranes and liposomes loaded with quercetin, respectively. The macrophage–liposome conjugation (MP-QT-NP) was fabricated via host–guest interactions between β-CD and ADA. The results showed that MP-QT-NP inhibited plaque inflammation [109]. Similarly, a drug delivery system was developed to encapsulate CD-based nanoparticles loaded with therapeutic compounds via macrophages, and this system ultimately attenuated inflammation in mice with atherosclerosis [110]. Novelly, β-CD with a ROS-responsive motif was linked to a two-photon aggregation-induced luminescent active group, and then prednisolone was loaded into its cavity via supramolecular interaction, thereby constructing a two-photon fluorophore–cyclodextrin/prednisolone complex which was then coated with nanomicelle. This composite nanomicelle enhanced anti-inflammatory activity by accumulating in atherosclerotic tissue through the damaged vascular endothelium [64]. A pH/ROS dual-sensitive nanoparticle was created by combining a pH-sensitive material with an oxidation-responsive material, both of which were simply synthesized by chemically functionalized β-CD. This nanosystem can be used as an effective and safe vehicle for the precise treatment of arterial restenosis and other inflammatory vascular diseases [65,111].

### 5.6. Applications of Cyclodextrin-Based Delivery Systems in Other Inflammatory Diseases

In the case of oral infections, liposomes grafted with CD can be used as a bioadhesive dual-drug nanocarrier, which can sustainably deliver multiple drugs to oral cells [112]. In neurodegenerative diseases, hydroxypropyl-β-cyclodextrin can be used as an intranasal delivery carrier of curcumin for the treatment of Alzheimer’s disease [113]. Additionally, a new epidural carrier consisting of microneedles and β-CD metal–organic frameworks was utilized to deliver drugs for the treatment of spinal cord injuries. With a high capacity for drug loading and the ability to penetrate the dura and control the release of drugs, this drug delivery system effectively reduced inflammation and improved neurological functions [66]. In particular, a delivery vehicle was framed to deliver drugs for the treatment of liver fibrosis by conjugating CD with vitamin A, using PEG2000 as a linker. This system attenuated liver inflammation and was a potential treatment for liver fibrosis [114].

## 6. Conclusions

In short, cyclodextrin-based nano-delivery systems have garnered significant attention in the field of smart drug delivery owing to their capacity to enhance drug bioavailability, enable site-specific targeted accumulation, prolong the systemic circulation duration, facilitate synergistic therapeutic outcomes, and exhibit superior biocompatibility profiles. Although cyclodextrin-based drug delivery systems demonstrate potential advantages in therapeutic applications, the current evaluation methodologies for these systems remain predominantly based on preclinical-stage experimental models, encompassing both in vitro analyses and animal-based in vivo studies, rather than comprehensive clinical validation. Translating these studies into the clinic remains a multi-dimensional challenge. At the biological level, interspecies variations in pharmacodynamics/pharmacokinetics make it difficult to extrapolate experimental data to clinical contexts. Notably, the differences in immunogenicity and metabolic enzyme expression patterns between rodents and humans may significantly affect targeting efficiency and toxicity thresholds. In addition, most of the existing studies focus on acute toxicity and lack systematic studies on the potential risk of CD accumulation in vivo, and long-term biosafety issues need to be addressed, focusing on the potential interference of CDs in lipid metabolic pathways, especially metabolic disorders that may be caused by the competitive binding of CDs to cholesterol in cell membranes. From a technical perspective, key challenges persist in optimizing encapsulation efficiency and stability, achieving sterilization process compatibility, and establishing standardized quality control protocols for large-scale production. Additionally, developing robust evaluation methodologies for the in vitro–in vivo correlation assessment of complex drug delivery systems requires further scientific advancement.

Future research can focus on the following directions: First, bio-inspired functionalized CD derivatives can be developed to improve their targeting and biological adaptability by simulating natural molecular structures. Second, organoids or organ-on-a-chip models can be established to simulate the complex physiological environment of the human body to optimize the design of the delivery systems. In addition, innovative applications of CDs in gene editing and cell therapy technologies can be explored. At the same time, it is necessary to strengthen the deep integration of computational simulation and experimental research and predict the interaction mechanism of drug–CD complexes through molecular dynamics simulation and artificial intelligence technology, so as to provide theoretical support for the theoretical design of intelligent delivery systems. Through interdisciplinary collaboration and the establishment of a translational medicine research system, CD-based drug delivery systems show potential to overcome the existing technical barriers and drive significant advancements in precision therapeutics.

## Figures and Tables

**Figure 1 pharmaceutics-17-00378-f001:**
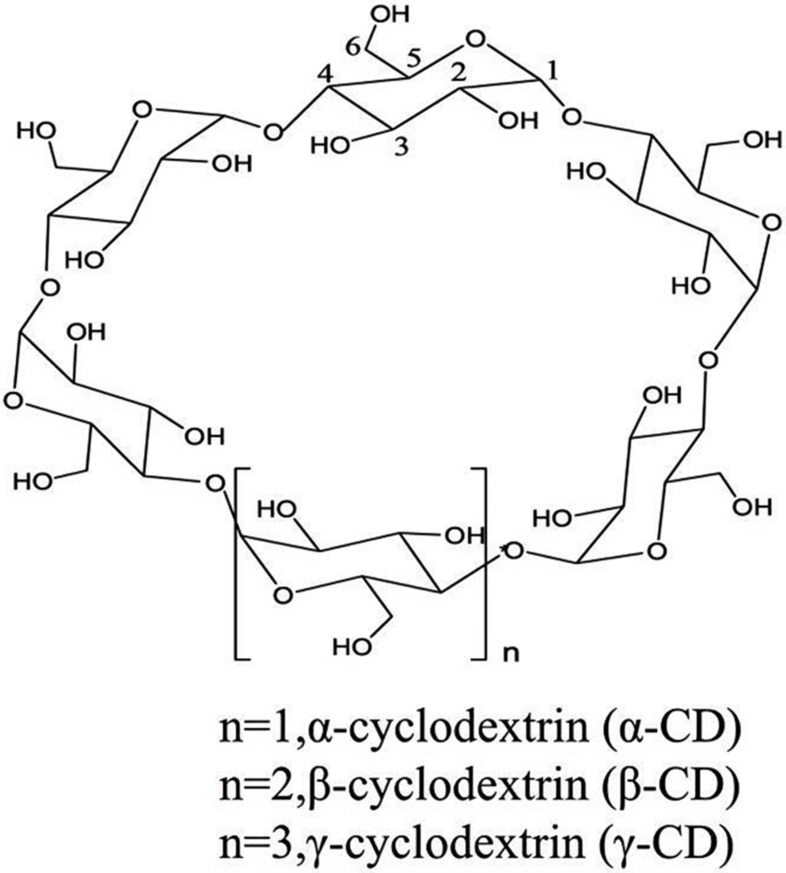
The chemical structure of cyclodextrins.

**Figure 2 pharmaceutics-17-00378-f002:**
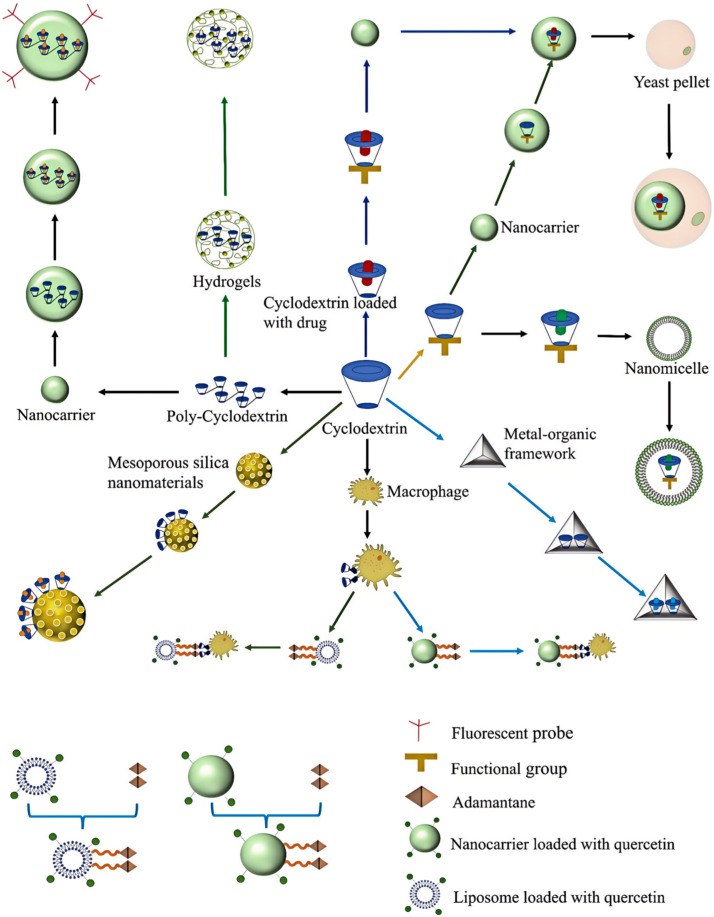
Assembly process of a nano-delivery system based on cyclodextrin. The figure is narrated clockwise from 12 o’clock. One route is that cyclodextrins are loaded with drugs, then modified with functional groups and finally loaded into nanocarriers. Another route is that cyclodextrins modified with functional groups are loaded into nanocarriers, which are then loaded with a drug. A third route is that cyclodextrins loaded with drugs are modified with functional groups, then loaded into a nanocarrier, and finally, this assembly is encapsulated by a yeast pellet. A fourth route is that cyclodextrins modified with functional groups load the drug, and finally, the complex is encapsulated by a nanomicelle. A fifth route is for a drug to be linked with a cyclodextrin-based metal–organic framework. A sixth route is to use cyclodextrin-modified macrophages crosslinked with ADA-modified nanocarriers carrying quercetin. A further route is to use cyclodextrin-modified macrophages crosslinked with ADA-modified liposomes carrying quercetin. Another route is that the drug is loaded in cyclodextrin-modified mesoporous silica nanoparticles. A further route is that a cyclodextrin-based nanoassembly is loaded with the drug and then crosslinked with a fluorescent probe. The final route is for the drug to be loaded onto poly-hydrogel containing cyclodextrin contact lenses.

**Figure 3 pharmaceutics-17-00378-f003:**
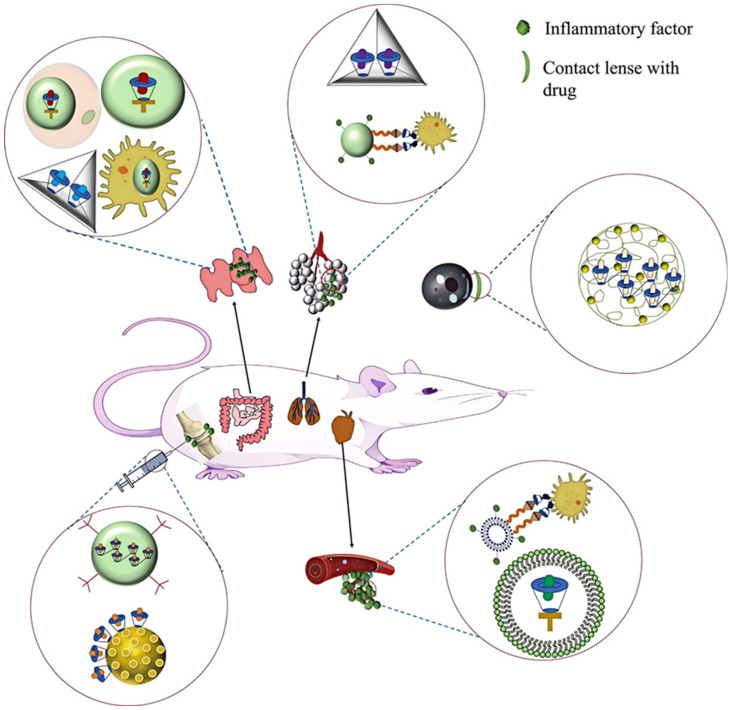
Application of cyclodextrin-based drug systems in inflammation-related diseases. There are four kinds of drug systems for inflammatory bowel disease. One is supramolecular nanoparticles encapsulated by a yeast pellet. The second is an inclusion complex coated by macrophages. The third is the crosslinking of drugs and cyclodextrin-based metal–organic frameworks. The fourth is a cyclodextrin-based nanosystem with PH/ROS sensitivity. There are two kinds of drug systems for respiratory diseases: one is cyclodextrin-based metal–organic framework-loaded drugs, and the other is cyclodextrin-modified macrophages crosslinked with ADA-modified nanocarriers carrying quercetin. One drug system for inflammatory eye disease is poly-hydrogel containing cyclodextrin contact lenses. There are two kinds of drug systems for vascular diseases. One is the crosslinking of cyclodextrin-modified macrophages with ADA-modified liposomes carrying quercetin. The second is the complex wrapped in nanomicelles. There are two kinds of drug systems for inflammatory joint disease. One is that the drugs are loaded on cyclodextrin-modified mesoporous silica nanoparticles. The other is a cyclodextrin-based nanoassembly loaded with drugs and crosslinked with fluorescent probes.

**Table 1 pharmaceutics-17-00378-t001:** Summary of the most important data on cyclodextrin-based delivery systems.

System	CD	APIs	Loading Capacity	Loading Efficiency	Drug Release Mechanism	In Vivo Studies	Ref
Polyanionic cyclodextrin supramolecular nanoparticles	Hepta-carboxyl-β-CD	HPTS	66%	-	Diffusive release	-	[49]
Targeted self-assembled supramolecular nanoparticles	Poly-β-CD	DOX and DTX	-	82.3% and 77.2%, respectively	pH change	-	[50]
GO-Phe-CD nanocarriers	β-CD	DOX	85.2%	78.7%	pH change	-	[51]
GO hybrid CD-based supramolecular hydrogels	α-CD	5-FU	-	-	NIR light-, temperature-, and pH-responsive release	-	[52]
Cyclodextrin inclusion micro-cocrystal	γ-CD	Asiatic acid	91.2%	11.4%	Diffusive release	Inflammatory factors and gene expression in lung tissue	[53]
Hydroxypropyl-β-cyclodextrin inclusion compound	HP-β-CD	Tetrandrine	-	93.28 ± 0.58%	Diffusive release	Pharmacokinetics, biodistribution, and efficacy	[54]
γ-cyclodextrin metal–organic frameworks	γ-CD	Luteolin	65%	-	Diffusive release	Pharmacokinetics, biodistribution, and efficacy	[55]
β-cyclodextrin inclusion complexes	β-CD	Andrographolide	9.61 ± 1.99%	63.92 ± 3.98%	Diffusive release	Anti-pneumonia efficacy and immune response	[56]
Cyclodextrin cationic polymer-based nanoassemblies	PolyCD	DCF	92%	100%	Diffusive release	-	[57]
Supramolecular nano-delivery systems	HP-β-CD	Curcumin	90.24% ± 1.49%	8.54 ± 0.08%	pH/ROS-responsive release	Biodistribution, anti-inflammatory antioxidant activities, and efficacy	[58]
H_2_O_2_-responsive covalent cyclodextrin frameworks	γ-CD	p-Hydroxybenzyl alcohol	-	-	H_2_O_2_-responsive release	Biodistribution, pharmacokinetics, and efficacy	[59]
A multiple-carbohydrate-based nanosystem	OX-CD	Dexamethasone	86.3 ± 1.2%	8.7 ± 0.1%	pH/ROS-responsive release	Biodistribution and therapeutic efficacy	[24]
pHEMA/β-CD contact lenses	β-CD	Puerarin	4~5%	18~30%	Diffusive release	Bioavailability and pharmacokinetics	[60]
pHEMA-co-β-CD	β-CD	Thiosemicarbazone	-	5%	Diffusive release	-	[61]
pHEMA-co-beta-CD hydrogels	β-CD	Hydrocortisone and acetazolamide	-	Less than 10%	Diffusive release	-	[62]
Chitosan/sulfobutylether-β-cyclodextrin-based nanoparticles	β-CD	Indomethacin	94.3%	3.32%	Diffusive release	Mucoadhesive, release, and permeation studies	[63]
Two-photon fluorophore–cyclodextrin/prednisolone complexes	β-CD	Prednisolone	93%	-	ROS-responsive release	Toxicity, pharmacokinetic study, and antiatherosclerosis activity	[64]
pH/ROS dual-responsive NPs	β-CD	Rapamycin	8.7 ± 0.4%	-	pH/ROS-responsive release	Targeting capability and acute toxicity	[65]
Microneedles and β-cyclodextrin metal–organic frameworks	β-CD	Methylprednisolone sodium succinate	60.49–72.66%	-	pH change	Biodistribution and therapeutic efficacy	[66]

APIs: active pharmaceutical ingredients; PLGA: Poly lactic-co-glycolic acid; GO: graphene oxide; Phe: L-plenylalanine; pHEMA: Poly (2-hydroxyethyl methacrylate).

## Data Availability

Data sharing is not applicable to this article as no datasets were generated or analyzed during the current study.

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
