# Peer review of "Applications of Cyclodextrin-Based Drug Delivery Systems in Inflammation-Related Diseases"

_pharmaceutics, 2025, doi:10.3390/pharmaceutics17030378_

Round 1
Reviewer 1 Report
Comments and Suggestions for Authors
The review provides a fairly comprehensive analysis of cyclodextrins.
I have some issue about section 3.2
1. do not use acronyms in the title
2. revise the English of the title
3. this section is superficial and it is not clear how it is related to the rest.
Author Response
Dear reviewers:
Thanks a lot for your comments to our manuscript titled “Applications of Cyclodextrin-based Drug Delivery Systems in Inflammation-related Diseases” (pharmaceutics-3452062). Those comments are very valuable and helpful for improving our manuscript. We have studied those comments carefully and revised our manuscript. We hope the revised version meet with approval. Revised portions are marked in red in revised manuscript. We tried our best to improve our manuscript and made some changes in the revised manuscript. These changes will not influence the content and framework of the paper. We did not list these changes but marked in red in the revised paper. We appreciate for editor and reviewers′ warm work earnestly, and hope that the revised manuscript will meet with approval.
The responses to the reviewers′ comments are following
The review provides a fairly comprehensive analysis of cyclodextrins. I have some issue about section 3.2.
Question 1: do not use acronyms in the title
Answer:
Thank you for your good advice. According to the second reviewer, we found that the section 3.2 is not consistent with the main thesis of the manuscript, and section 3.2 has been deleted.
Question 2: .revise the English of the title
Answer:
Thank you for your advice. According to the second reviewer, we found that the section 3.2 is not consistent with the main thesis of the manuscript, and section 3.2 has been deleted.
Question 3: this section is superficial and it is not clear how it is related to the rest.
Answer:
Thank you for your valuable advice. According to the second reviewer, section 3.2 cannot be related to the rest of the manuscript, we have deleted it for the sake of the coherence of the manuscript
Reviewer 2 Report
Comments and Suggestions for Authors
Dear Authors, please find the below suggestions to improve the manuscript quality.
- The introduction lacks a strong research question or justification for why this review is necessary. The authors provide a generic overview of inflammation and cyclodextrins in drug delivery without explaining what gaps in knowledge exist.
- Structure and Properties of Cyclodextrins: This is excessive background information on cyclodextrins that does not contribute to the main focus. The authors should streamline the discussion and focus on aspects directly related to drug delivery applications.
- 2 Delivery Systems Modified based on Cyclodextrins Penetrate BBB: Is this not a type of chemical modification? The subdivisions in this section do not align with the flow of the manuscript.
- Nano Delivery Systems based on Cyclodextrins: What are the advantages and limitations of these systems? The author does focus on case studies but should also mention the potential limitations.
- In Conclusion, The manuscript fails to critically address the limitations of cyclodextrin-based drug delivery, such as stability, metabolism, and regulatory concerns.
- In Table 1, What should be the ideal number for Loading capacity and efficiency? Please find out and provide the information in the Notes below the Table.
- Please add this article in references https://www.sciencedirect.com/science/article/abs/pii/S0167732224016714 ,2024; and a research article on cyclodextrin-Ibuprofen complexation Pharmaceutics2023, 15(9), 2203; https://doi.org/10.3390/pharmaceutics15092203.
Author Response
Dear reviewers:
Thanks a lot for your comments to our manuscript titled “Applications of Cyclodextrin-based Drug Delivery Systems in Inflammation-related Diseases” (pharmaceutics-3452062). Those comments are very valuable and helpful for improving our manuscript. We have studied those comments carefully and revised our manuscript. We hope the revised version meet with approval. Revised portions are marked in red in revised manuscript. We tried our best to improve our manuscript and made some changes in the revised manuscript. These changes will not influence the content and framework of the paper. We did not list these changes but marked in red in the revised paper. We appreciate for editor and reviewers′ warm work earnestly, and hope that the revised manuscript will meet with approval.
The responses to the reviewers′ comments are following
Reviewer #2:
Dear Authors, please find the below suggestions to improve the manuscript quality.
Question 1: The introduction lacks a strong research question or justification for why this review is necessary. The authors provide a generic overview of inflammation and cyclodextrins in drug delivery without explaining what gaps in knowledge exist.
Answer:
We sincerely appreciate your valuable feedback. We have revised the "introduction" part of the manuscrip. The changed part has been highlighted in red, and the deleted part is not shown. The changes in the revised manuscript as follows:
“⋯⋯ As a traditional drug carrier, Cyclodextrins (CDs) are worth building due to their unique annular cavity structure, modified surface hydroxyl groups, good biocompatibility, inclusion complex ability and water solubility [17-20]. Cyclodextrin-based nano-delivery systems have garnered significant attention in contemporary pharmaceutical research owing to their capacity to enhance drug bioavailability, enable site-specific targeted accumulation, prolong systemic circulation duration, facilitate synergistic therapeutic outcomes, and exhibit superior biocompatibility profiles [21-23]. It is worth noting that cyclodextrin-based nano-delivery systems can achieve targeted delivery and controlled release of drugs in the treatment of inflammation-related diseases, thus ameliorating inflammation. For example, Yuan Wang et al. developed a cyclodextrin-based nano-delivery system with pH/ROS dual responsibility for inflammatory bowel disease [24]. Cyclodextrin-based drug delivery systems show great potential in inflammation-related diseases. However, few studies systematically review their design strategies and application advancements.
In this review, we summarized the structure and chemical modification of cyclodextrins. Second, the application of cyclodextrin-based drug delivery systems in inflammation-related diseases was summarized. Finally, the current obstacles and future directions of cyclodextrin-based delivery systems were discussed.”
Question 2: Structure and Properties of Cyclodextrins: This is excessive background information on cyclodextrins that does not contribute to the main focus.
Answer:
Thank you for your valuable advice. According to your suggestion, we have deleted the content that is not related to the main thesis of the manuscript in the section "Structure and Properties of Cyclodextrins". A diagram of the chemical structure of cyclodextrins has been added. This section was adjusted in the revised manuscript as follows:
“Cyclodextrins (CDs) are produced by degradation of starch under the action of CD glucosyltransferase, which is a class of cyclic oligosaccharides formed by 5 or more α-D-glucopyranose units linked by α-(1,4) glucoside bonds. Common cyclodextrins are α-cyclodextrin (α-CD), β-cyclodextrin (β-CD), and γ-cyclodextrin (γ-CD) containing 6, 7, and 8 glucose subunits, respectively [25-27]. β-CD is widely used in pharmaceutical excipients because of its suitable cavity size, effective drug load, and relatively low cost [28]. Cyclodextrin is a physically and chemically stable macromolecule with a truncated cone or torus structure with a hydrophilic external structure and a hydrophobic internal cavity [29-32]. Such a structural feature presents a unique dual property that interacts with both lipophilic and hydrophilic compounds [33]. Importantly, their hydrophobic internal cavity can trap or encapsulate drugs (guests), thus forming non-covalently bound host-guest inclusion complexes to improve the physical, chemical and biological properties of the guest molecules [32, 34-36]. Fig 2 showed the chemical structure of cyclodextrin”.

Fig 2 The chemical structure of cyclodextrin
Question 3:The authors should streamline the discussion and focus on aspects directly related to drug delivery applications.
Answer:
Thank you for your reminder and suggestion. According to your suggestions, we have rewritten the section "Conclusion" of the manuscript. The changed part has been highlighted in red, and the deleted part is not shown. The changes in the revised manuscript are as follows:
“In short, cyclodextrin-based nano delivery systems have garnered significant attention in the field of smart drug delivery owing to their capacity to enhance drug bioavailability, enable site-specific targeted accumulation, prolong systemic circulation duration, facilitate synergistic therapeutic outcomes, and exhibit superior biocompatibility profiles. Although cyclodextrin-based drug delivery systems demonstrate considerable potential advantages in therapeutic applications, current evaluation methodologies for these systems remain predominantly based on preclinical-stage experimental models, encompassing both in vitro analyses and animal-based in vivo studies, rather than comprehensive clinical validation. Translating these studies into the clinic remains a multi-dimensional challenge. At the biological level, interspecies variations in pharmacodynamics/pharmacokinetics make it difficult for extrapolating experimental data to clinical contexts. Notably, the differences in immunogenicity and metabolic enzyme expression patterns between rodents and humans may significantly affect targeting efficiency and toxicity thresholds. In addition, most of the existing studies focus on acute toxicity and lack systematic studies on the potential risk of CDs accumulation in vivo, and long-term biosafety issues need to focus on the potential interference of CDs on lipid metabolic pathways, especially the metabolic disorders that may be caused by the competitive binding of CDs to cholesterol in cell membranes. From a technical perspective, key challenges persist in optimizing encapsulation efficiency and stability, achieving sterilization process compatibility, and establishing standardized quality control protocols for large-scale production. Additionally, developing robust evaluation methodologies for in vitro-in vivo correlation assessment of complex drug delivery systems requires further scientific advancement.
Future research can focus on the following directions. First, bio-inspired functionalized CDs derivatives can be developed to improve their targeting and biological adaptability by simulating natural molecular structures. Secondly, organoids or organ-on-a-chip models can be established to simulate the complex physiological environment of the human body to optimize the design of the delivery systems. In addition, innovative applications of CDs in gene editing and cell therapy technologies can be explored. At the same time, it is necessary to strengthen the deep integration of computational simulation and experimental research, and predict the interaction mechanism of drug-CD complex through molecular dynamics simulation and artificial intelligence technology, so as to provide theoretical support for the theoretical design of intelligent delivery systems. Through interdisciplinary collaboration and establishment of a translational medicine research system, the CD-based drug delivery systems show potential to overcome the existing technical barriers and drive significant advancements in precision therapeutics.”
Question 4:Delivery Systems Modified based on Cyclodextrins Penetrate BBB: Is this not a type of chemical modification? The subdivisions in this section do not align with the flow of the manuscript.
Answer:
Thank you for your valuable advice. We found that the section 3.2 is not consistent with the main thesis of the manuscript, and section 3.2 has been deleted
Question 5:Nano Delivery Systems based on Cyclodextrins: What are the advantages and limitations of these systems? The author does focus on case studies but should also mention the potential limitations.
In Conclusion, The manuscript fails to critically address the limitations of cyclodextrin-based drug delivery, such as stability, metabolism, and regulatory concerns.
Answer:
Thank you for your valuable advice. Through a comprehensive literature review, we have supplemented the content about the constraints and challenges associated with cyclodextrin-based nano delivery systems. The changed part has been highlighted in red. The changes in the revised manuscript are as follows:
“4.1 Polymeric Nanosystems based on Cyclodextrins
CD-based polymeric nanoparticles exhibit unique advantages in the field of drug delivery[49, 50]. However, the nanoparticles have inherent limitations, such as low encapsulation rates and drug loading. Nanoparticles aided by CDs yield a novel drug delivery system with the benefits of both components: the CDs offer improved water solubility and drug loading, whereas the NPs provide targeted drug delivery. The delivery systems have cyclodextrin casting outer shells, while the core of the polymeric nanoparticles is composed of natural or synthetic polymer. The drugs can either be loaded into the core of the polymer nanoparticle core or be conjugated with the cyclodextrin in the outer shell[19, 49].Many studies have shown the importance of CD-based polymer nanoparticle systems for realizing stimuli-responsive drug delivery systems. An efficient delivery system was developed by combining CDs and polymer nanoparticles (PNPs). When CDs were improving their efficiency of drug loading, PNPs can be utilized to synergize this intelligent delivery system in response to endogenous stimuli (such as pH, enzymes, temperature) or exogenous stimuli (such as magnetic, light, and ultrasound), thereby accelerating the release of drugs [51]. However, this composite system still faces some limitations in practice. For example, the preparation of polymer nanoparticles is a complex process that is highly sensitive to solvent selection, PH and temperature[52, 53].
Supramolecular self-assembled nanocarriers based on CDs also belong to polymeric nanosystems, and their performance can be optimized by molecular functional strategies. Studies have shown that structural modification of β-CD with different functional groups (such as amantadane, azobenzene, etc.) or polymers (such as PEG, PEI, etc.) can improve the solubility of the therapeutic molecules and the stability of the inclusion complexes and regulatethe interface affinity of the macromolecular self-assemble systems in the biological environment, thus overcoming the excessive aggregation problems of macromolecular self-assembly systems[54, 55]. For example, amantadine and amphiphilic β-CD formed a stable molecular aggregate that was used to deliver HPTS (8-hydroxypyren-1,3,6-trisulfonic acid). After 360min, the release rate of HPTS encapsulated in supramolecular nanoparticles in water was reported to be 8-fold lower than that of free HTPS (80%) [55]. In addition, a nanocarrier was constructed for targeted delivery and combination chemotherapy, utilizing the target specificity of aptamers (oligonucleotides) and self-assembleing capabilities of CD-based supramolecules. The results showed that the self-assembled nanocarriers based on CDs provide an efficient platform for targeted drug delivery systems with synergistic antitumor activity[56]. However, CD-based supramolecular self-assembled nanosystems still face technical bottlenecks in their applications. First, the instability in structure of the host-guest complex under acid-base conditions, which leads to some covalent/non-covalent assemblies prone to dissociation. Secondly, supramolecular self-assembled nanosystems based on CDs are extremely sensitive to the parameters in process (solvent polarity, concentration gradient, stirring speed, etc.). However, the vulnerability can be weakened by CD-polymer conjugation, which has better loading rate and stability[57, 58].
4.2 Graphene Derivatives Delivery Systems modified with Cyclodextrins
In drug delivery systems, graphene-derived nanoparticles were introduced to increase drug loading and control drug release by stimuli (e.g., alternating magnetic fields, changing pH or temperature) [59]. Among them, graphene oxide (GO) has become a research hotspot in the field of nanomedicine delivery due to its high specific surface area and abundant surface functional groups[60]. The CD-modified graphene oxide (GO) nanomedicine delivery systems take advantage of the cavity structure of CD and the stimuli-responsive release ability of GO. The delivery systems have a high drug loading capacity and loading efficiency, enabling targeted delivery and controlled release. Graphene oxide (GO) has been a hot topic of research [60]. Interestingly, a CD-modified nanocarrier based on GO was synthesized for cancer therapy using L-phenylalanine as a link, the load efficiency and load capacity of this system for DOX were 78.7% and 85%, respectively. After 72h, only 12% of DOX was slowly released at pH 7.4, while up to 40% of DOX was released at pH 5.3 under acidic conditions [61]. In addition, a supramolecular hydrogel consisting of GO and α-CD was designed. This supramolecular hydrogel increased drug loading through host-guest interactions and released the anticancer drug 5-fluorouracil in response to near-infrared light (NIR), temperature, and pH. In this system, GO was used as a core material for additional cross-linking to confer thermal stability. Meanwhile, GO absorbed near-infrared light and converted part of the heat into local heat to trigger the gel to sol transition[62]. However, there are still some limitations in the application of CD-modified graphene derivative delivery systems. First, although the cytotoxicity possessed by GO can partially be reduced by CDs modification, the long-term biosafety of this composite system is not yet clear[63, 64]. In addition, most of the existing methods for the preparation of CD-modified GO rely on sophisticated chemical reactions, which makes it difficult to convert to large-scale production[65].
4.3 Cyclodextrin-based Inorganic Nanoparticle Systems
Cyclodextrin-based inorganic nanoparticle delivery systems have attracted much attention due to their unique advantages in the field of targeted drug delivery and sustained release. In this system, commoninorganic nanomaterials mainly include mesoporous silica nanoparticles, plasmonic nanoparticles, magnetic nanoparticles, and quantum dots. Among them, mesoporous silica nanoparticles exhibit significant advantages in targeted delivery due to its internal pores with a large surface area and tunable mesoporous structure[66].Since mesoporous silica nanoparticles have no ability to respond to stimuli, they must be chemically modified using polymers or supramolecular structures to control release of drugs. On one hand, the derivatives of CDs were used as gatekeepers to control the release of drugs from premature release. Furthermore, CDs were loaded with drugs into mesoporous silica matrices and responded to external stimulus signals through host-guest interactions [67]. On the other hand, the derivatives of CDs contributed to the synthesis of these nanoparticles as reductants and stabilizers and increased the loading capacity of the drugs [68].
However, despite the potential of this system in terms of drug loading, biocompatibility, and targeting, the systems have several limitations that restrict its clinical translation and application. Firstly, silica nanoparticles can achieve drug loading through physical adsorption or chemical bonding through a mesoporous structure, however, their structure is unstable in complex physiological environment (such as pH fluctuations) in vivo[69]. Secondly, silica nanoparticles are easily adsorbed by plasma proteins to form a "protein corona" following systemic administration, thus obscuring the targeted ligands and alters their biological distribution[70, 71]. Furthermore, the biodegradability of mesoporous silica materials has not been fully elucidated in vivo, which may bring some long-term physiological toxicity[72]. What's more, long-term retention of high doses of mesoporous silica can even cause organ fibrosis[73].
4.4 Cyclodextrin-Liposome Systems
Liposomes are small spherical vesicles that are synthesized and are composed of one or more lipid bilayers that enclose the inner aqueous space [74]. The structure makes liposomes suitable for encapsulating hydrophobic, hydrophilic, and amphiphilic substances [75]. Different strategies are developed for the binding of CDs to liposomes, one of which is called "drug-in CD-in liposome systems". The inclusion complexes are encapsulated in the aqueous nuclei of different liposomes. The dual-loading nano-delivery system increases the solubility of the active pharmaceutical ingredient and the stability of the vesicles. Compared to single-loading mode, this dual-loading technology enabled either rapid or prolonged release of the active pharmaceutical ingredient [76, 77]. Drug-in CD-in liposome formulations have been shown to improve the encapsulation efficiency and loading ratio[78].
However, there may be some issues that need to be considered for composite delivery systems of CDs and liposomes. First, Drug-in CD-in liposome formulations should take into account the effect of CDs on membranes of liposomes[79]. In short, CDs molecules present in the aqueous phase space of the liposome will interact with phospholipids on the liposome membrane, thus affecting the stability of CDs and changing its fluidity and permeability[80, 81]. Furthermore, lipids exhibit hemolytic activity, as evidenced by in vitro experimental studies[82]. Accordingly, the potential toxicological implications associated with CD-liposome system need to be investigated.
4.5 Other Nanodelivery Systems based on Cyclodextrins
In addition to the nanomaterials mentioned above, CDs or their derivatives can also interact with metal-organic frameworks (MOFs) [83], Janus nanoparticles [84], nanofibers [85], or nanomicelles to build novel drug delivery systems.
Question 6:In Table 1, What should be the ideal number for Loading capacity and efficiency? Please find out and provide the information in the Notes below the Table.
Answer:
Agreed. Thank you very much for your suggestion. After literature review, most of the articles only show the drug loading capacity and efficiency, while the ideal number for loading capacity and efficiency is not elucidated. Could you give us some guidelines?
Reviewer 3 Report
Comments and Suggestions for Authors
Although Dai and colleagues provide a commendable review on cyclodextrin-based anti-inflammatory drug delivery, the current version falls well short of publication standards. The manuscript requires extensive revisions because of the repetitive content and poor language, making it challenging for readers to extract essential information, even from the abstract.
The closing two sentences lack coherence and are especially difficult to comprehend.
Furthermore, despite listing eleven authors, the content does not substantiate such a number. It is evident that at least half of the authors assert their perceived rights; however, their meaningful input in the text is not readily identifiable.
Moreover, the keywords are nothing more than echoes of the title, which is wholly inadequate and undermines the manuscript's quality.
Significantly, there are also inaccuracies, such as the claim surrounding the mass production of Hydroxypropyl-γ-CD. Although (2-hydroxy)propylation does enhance the solubility of parent cyclodextrins indeed — especially HPβCD for βCD — by mitigating aggregation that causes precipitation or low aqueous solubility, the reality of mass production is primarily associated with HPβCD, sulfobutyl βCD, or the methylated βCD, though this latter remain inappropriate for parenteral administration.
The authors fail to draw a critical distinction between accepted formulations and merely conjectural developments, which significantly diminishes the reliability and academic rigor.
Due to the weakness of clarity and precision, the authors must focus on improving the manuscript language and content to an acceptable standard.
The journal abbreviations should contain dot(s).
Comments on the Quality of English LanguageRevision of a native English scientist is mandatory.
Author Response
Dear reviewers:
Thanks a lot for your comments to our manuscript titled “Applications of Cyclodextrin-based Drug Delivery Systems in Inflammation-related Diseases” (pharmaceutics-3452062). Those comments are very valuable and helpful for improving our manuscript. We have studied those comments carefully and revised our manuscript. We hope the revised version meet with approval. Revised portions are marked in red in revised manuscript. We tried our best to improve our manuscript and made some changes in the revised manuscript. These changes will not influence the content and framework of the paper. We did not list these changes but marked in red in the revised paper. We appreciate for editor and reviewers′ warm work earnestly, and hope that the revised manuscript will meet with approval.
The responses to the reviewers′ comments are following
Reviewer #3:
Question 1: Although Dai and colleagues provide a commendable review on cyclodextrin-based anti-inflammatory drug delivery, the current version falls well short of publication standards. The manuscript requires extensive revisions because of the repetitive content and poor language, making it challenging for readers to extract essential information, even from the abstract.
Answer:
My sincerely appreciated the experts for your advice. In terms of content, the text has been comprehensively restructured by removing of redundant sections and extraneous material unrelated to the themes. In terms of language, the language has been modified by native-speaking in revised manuscript. We did not list these changes but marked in red in the revised manuscript.
Question 2: The closing two sentences lack coherence and are especially difficult to comprehend.
Answer:
Thank you for your reminder and suggestion. According to your suggestion, we have rewritten the "Conclusion" of the manuscript, the changed part has been marked in red. The changes were followed.
“In short, cyclodextrin-based nano delivery systems have garnered significant attention in the field of smart drug delivery owing to their capacity to enhance drug bioavailability, enable site-specific targeted accumulation, prolong systemic circulation duration, facilitate synergistic therapeutic outcomes, and exhibit superior biocompatibility profiles. Although cyclodextrin-based drug delivery systems demonstrate considerable potential advantages in therapeutic applications, current evaluation methodologies for these systems remain predominantly based on preclinical-stage experimental models, encompassing both in vitro analyses and animal-based in vivo studies, rather than comprehensive clinical validation. Translating these studies into the clinic remains a multi-dimensional challenge. At the biological level, interspecies variations in pharmacodynamics/pharmacokinetics make it difficult for extrapolating experimental data to clinical contexts. Notably, the differences in immunogenicity and metabolic enzyme expression patterns between rodents and humans may significantly affect targeting efficiency and toxicity thresholds. In addition, most of the existing studies focus on acute toxicity and lack systematic studies on the potential risk of CDs accumulation in vivo, and long-term biosafety issues need to focus on the potential interference of CDs on lipid metabolic pathways, especially the metabolic disorders that may be caused by the competitive binding of CDs to cholesterol in cell membranes. From a technical perspective, key challenges persist in optimizing encapsulation efficiency and stability, achieving sterilization process compatibility, and establishing standardized quality control protocols for large-scale production. Additionally, developing robust evaluation methodologies for in vitro-in vivo correlation assessment of complex drug delivery systems requires further scientific advancement.
Future research can focus on the following directions. First, bio-inspired functionalized CDs derivatives can be developed to improve their targeting and biological adaptability by simulating natural molecular structures. Secondly, organoids or organ-on-a-chip models can be established to simulate the complex physiological environment of the human body to optimize the design of the delivery systems. In addition, innovative applications of CDs in gene editing and cell therapy technologies can be explored. At the same time, it is necessary to strengthen the deep integration of computational simulation and experimental research, and predict the interaction mechanism of drug-CD complex through molecular dynamics simulation and artificial intelligence technology, so as to provide theoretical support for the theoretical design of intelligent delivery systems. Through interdisciplinary collaboration and establishment of a translational medicine research system, the CD-based drug delivery systems show potential to overcome the existing technical barriers and drive significant advancements in precision therapeutics.”
Question 3: Furthermore, despite listing eleven authors, the content does not substantiate such a number. It is evident that at least half of the authors assert their perceived rights; however, their meaningful input in the text is not readily identifiable.
Answer:
Thank you for your suggestion. We have made adjustments to the co-authors and the adjustments were agreed by all authors. Authors who made small contributions to the review were placed in the acknowledgements section of the manuscript. In the acknowledgments section, we would like to thank the author Yun Zhu, Hailian Zhu for their help in data collection and Lingyu Li, Bifeng He for their help in writing of the manuscript. In addition, the munuscript was commissioned, and it was required to complete in a short time, so more people participated in it.
Comments 4: Moreover, the keywords are nothing more than echoes of the title, which is wholly inadequate and undermines the manuscript's quality.
Answer:
We sincerely appreciate your astute observation. We have revised the keywords in revised manuscript as followed:
“Keywords: cyclodextrin; drug delivery system; nano-delivery; inflammation;”
Question 5: Significantly, there are also inaccuracies, such as the claim surrounding the mass production of Hydroxypropyl-γ-CD. Although (2-hydroxy)propylation does enhance the solubility of parent cyclodextrins indeed — especially HPβCD for βCD — by mitigating aggregation that causes precipitation or low aqueous solubility, the reality of mass production is primarily associated with HPβCD, sulfobutyl βCD, or the methylated βCD, though this latter remain inappropriate for parenteral administration.
Answer:
Thank you very much for your professional advice, which makes us deeply feel that the reviewer has profound professional knowledge. The content has been comprehensively restructured by removing of redundant sections and extraneous material unrelated to the themes in revised munuscript. In addition, we carefully read the contents of all the references in order to make our expression as consistent as possible with that of the original expression. We hope that the revised manuscript will meet with approval. However, if reviewers feel that it is still not clear enough, we would like your guidance to make our article more perfect. Thank you very much for your advice.
Here, "Hydroxypropyl-γ-CD has lower aggregation than unmodified γ-CD and is used for mass production" is replaced by "Hydroxypropyl-γ-CD has lower aggregation than unmodified γ-CD" in revised munuscript.
Question 6: The authors fail to draw a critical distinction between accepted formulations and merely conjectural developments, which significantly diminishes the reliability and academic rigor. Due to the weakness of clarity and precision, the authors must focus on improving the manuscript language and content to an acceptable standard.
Answer:
We sincerely appreciate your critical guidance. I'm very sorry for our inappropriate expression. We realized the manuscript's insufficient demarcation between fact and speculation, which led to inconsistencies between the Abstract and Section 5. Following a thorough re-examination of the manuscript, and the content of the Abstract is modified in the revised manuscript. The modified content of the Abstract is following:
“Abstract: Currently, inflammation diseases have become one of the leading causes of mortality worldwide. The therapeutic drugs for inflammation are mainly steroidal and non-steroidal anti-inflammatory drugs. However, the use of these anti-inflammatory drugs in a prolonged period is prone to cause serious side effects. Accordingly, it is particularly critical to design an intelligent target-specific drug delivery system to control the release of drugs in order to mitigate the side effects of anti-inflammatory drugs without limiting their activity. Meanwhile, cyclodextrin-based nano-delivery systems have garnered significant attention in contemporary pharmaceutical research owing to their capacity to enhance drug bioavailability, enable site-specific targeted accumulation, prolong systemic circulation duration, facilitate synergistic therapeutic outcomes, and exhibit superior biocompatibility profiles. It is worth noting that cyclodextrin-based drug delivery systems show great potential in inflammation-related diseases. However, few studies systematically review their design strategies and application advancements. Here, we summarize the structural and chemical modification strategies of cyclodextrins, as well as cyclodextrin-based drug delivery systems and their applications in inflammation-related diseases. In summary, the aim is to provide a bit of insight into the development of cyclodextrin-based drug delivery systems for inflammation-related diseases.”
Question 7: The journal abbreviations should contain dot(s).
Answer:
Thank you very much for your advice. We added a dot after the magazine abbreviation, Thank you.
Question 8:Comments on the Quality of English Language.
Revision of a native English scientist is mandatory.
Answer:
We are so sorry for our mediocre English. We had corrected the linguistic errors in the revised manuscript according to a native English speaker familiar with scientific writing. We hope the correction meet with approval.
Thanks again for your perfect comments to our manuscript.
Reviewer 4 Report
Comments and Suggestions for Authors
Please consider the following when revising the manuscript:
- Add more figures to improve the readability and understanding of the manuscript.
- Expand Section 5 to include data and figures from the studies cited and critical evaluation.
- Figure 1 needs modification and its legend needs to be more descriptive.
Author Response
Dear reviewers:
Thanks a lot for your comments to our manuscript titled “Applications of Cyclodextrin-based Drug Delivery Systems in Inflammation-related Diseases” (pharmaceutics-3452062). Those comments are very valuable and helpful for improving our manuscript. We have studied those comments carefully and revised our manuscript. We hope the revised version meet with approval. Revised portions are marked in red in revised manuscript. We tried our best to improve our manuscript and made some changes in the revised manuscript. These changes will not influence the content and framework of the paper. We did not list these changes but marked in red in the revised paper. We appreciate for editor and reviewers′ warm work earnestly, and hope that the revised manuscript will meet with approval.
The responses to the reviewers′ comments are following
Reviewer #4:
Please consider the following when revising the manuscript:
Question 1: Add more figures to improve the readability and understanding of the manuscript.
Answer:
Thanks for your advice. After reviewing the text, we have added " Graphical abstract ", " The chemical structure of cyclodextrins ", " Assembly process of a nano-delivery system based on cyclodextrin ", " Application of cyclodextrin-based drug systems in inflammation- related diseases ". The figures were shown below

Fig 1 Graphical abstract

Fig 2 The chemical structure of cyclodextrins

Fig3 Assembly process of a nano-delivery system based on cyclodextrin
The figure is narrated clockwise from 12 o 'clock. One route is that cyclodextrins are loaded with drugs, then modified by functional groups and finally loaded into nanocarriers; One route is that cyclodextrins modified with functional groups are loaded into nanocarriers, which are then loaded with a drug; One route is that cyclodextrins loaded with drugs are modified by functional groups, then loaded with a nanocarrier, and finally this assembly is encapsulated by a yeast pellet; One route is cyclodextrins modified with functional groups load the drug, and finally the complex is encapsulated by nanomicelle; One route is for a drug to be linked with a cyclodextrin-based metal-organic framwork; One route is to use cyclodextrin-modified macrophages crosslinked with ADA-modified nanocarriers carrying quercetin; One route is to use cyclodextrin-modified macrophages crosslinked with ADA-modified liposomes carrying quercetin; One route is that the drug is loaded in cyclodextrin-modified mesoporous silica nanoparticles; One route is that cyclodextrin-based nanoassembly loaded with the drug and then crosslinked with a fluorescent probe; One route is for the drug to be loaded onto poly hydrogel containing cyclodextrin contact lenses.

Fig. 4 Application of cyclodextrin-based drug systems in inflammation- related diseases
There are four kinds of drug systems for inflammatory bowel disease. One is supramolecular nanoparticles were encapsulated by a yeast pellet; The second is the inclusion complex was coated by macrophage. The third is the cross-linking of drugs and cyclodextrin-based metal-organic frameworks;The fourth is a cyclodextrin-based nanosystem with PH/ROS sensitivity. There are two kinds of drugs systems for respiratory diseases, one is cyclodextrin-based metal-organic frameworks loaded drugs; Second, cyclodextrin-modified macrophages cross-linked with ADA-modified nanocarriers carrying quercetin. One drug system for the inflammatory eye disease is poly hydrogel containing cyclodextrin contact lenses. There are two kinds of drugs systems for vascular diseases. One is the cross-linking of cyclodextrin-modified macrophages with ADA modified liposomes carrying quercetin. The second is the complex was wrapped by nanomicelles. There are two kinds of drugs systems for inflammatory joint disease. One is that the drugs are loaded on cyclodextrin-modified mesoporous silica nanoparticles. One is a cyclodextrin-based nanoassembly is loaded with drugs and cross-linked with fluorescent probes
Question 2: Expand Section 5 to include data and figures from the studies cited and critical evaluation.
Answer:
Thank you very much for your advice. Two diagram illustrating Part 5 have been added, as shown in the figure above. As for the data you mentioned that needs to be expanded, we understand that important data on drug delivery systems related to these experimental studies have been presented in Table 1, and you mean to describe the content again, is that right? Can you give us more details?
Question 3:Figure 1 needs modification and its legend needs to be more descriptive.
Answer:
Thank you for your advice. The diagram you mentioned has been refined, the legend have been added in revised munuscript. As shown in the following picture:

Fig3 Assembly process of a nano-delivery system based on cyclodextrin
The figure is narrated clockwise from 12 o 'clock. One route is that cyclodextrins are loaded with drugs, then modified by functional groups and finally loaded into nanocarriers; One route is that cyclodextrins modified with functional groups are loaded into nanocarriers, which are then loaded with a drug; One route is that cyclodextrins loaded with drugs are modified by functional groups, then loaded with a nanocarrier, and finally this assembly is encapsulated by a yeast pellet; One route is cyclodextrins modified with functional groups load the drug, and finally the complex is encapsulated by nanomicelle; One route is for a drug to be linked with a cyclodextrin-based metal-organic framwork; One route is to use cyclodextrin-modified macrophages crosslinked with ADA-modified nanocarriers carrying quercetin; One route is to use cyclodextrin-modified macrophages crosslinked with ADA-modified liposomes carrying quercetin; One route is that the drug is loaded in cyclodextrin-modified mesoporous silica nanoparticles; One route is that cyclodextrin-based nanoassembly loaded with the drug and then crosslinked with a fluorescent probe; One route is for the drug to be loaded onto poly hydrogel containing cyclodextrin contact lenses.
Round 2
Reviewer 2 Report
Comments and Suggestions for Authors
The revised manuscript is satisfactory.
Reviewer 3 Report
Comments and Suggestions for Authors
The authors have significantly improved their manuscript quality and transformed it into a valuable paper. Now it is suitable for publication.